# Transcriptional regulators of the *Golli/myelin basic protein* locus integrate additive and stealth activities

Hooman Bagheri[1], Hana Friedman[1], Katherine A. Siminovitch[2,3,4], Alan C. Peterson[1,5,6]*

1 Department of Human Genetics, McGill University, Montreal, Quebec, Canada, 2 Department of Medicine, University of Toronto, Toronto, Ontario, Canada, 3 Department of Immunology, University of Toronto, Toronto, Ontario, Canada, 4 Mount Sinai Hospital, Lunenfeld-Tanenbaum and Toronto General Hospital Research Institutes, Toronto, Ontario, Canada, 5 Gerald Bronfman Department of Oncology, McGill University, Montreal, Quebec, Canada, 6 Department of Neurology and Neurosurgery, McGill University, Montreal, Quebec, Canada

* alan.peterson@mcgill.ca

**Data Availability Statement:** All relevant data are within the manuscript and its Supporting Information files.

## Abstract

Myelin is composed of plasma membrane spirally wrapped around axons and compacted into dense sheaths by myelin-associated proteins. Myelin is elaborated by neuroepithelial derived oligodendrocytes in the central nervous system (CNS) and by neural crest derived Schwann cells in the peripheral nervous system (PNS). While some myelin proteins accumulate in only one lineage, *myelin basic protein* (*Mbp*) is expressed in both. Overlapping the *Mbp* gene is *Golli*, a transcriptional unit that is expressed widely both within and beyond the nervous system. A super-enhancer domain within the *Golli/Mbp* locus contains multiple enhancers shown previously to drive reporter construct expression specifically in oligodendrocytes or Schwann cells. In order to determine the contribution of each enhancer to the *Golli/Mbp* expression program, and to reveal if functional interactions occur among them, we derived mouse lines in which they were deleted, either singly or in different combinations, and relative mRNA accumulation was measured at key stages of early development and at maturity. Although super-enhancers have been shown previously to facilitate interaction among their component enhancers, the enhancers investigated here demonstrated largely additive relationships. However, enhancers demonstrating autonomous activity strictly in one lineage, when missing, were found to significantly reduce output in the other, thus revealing cryptic "stealth" activity. Further, in the absence of a key oligodendrocyte enhancer, *Golli* accumulation was markedly and uniformly attenuated in all cell types investigated. Our observations suggest a model in which enhancer-mediated DNA-looping and potential super-enhancer properties underlie *Golli/Mbp* regulatory organization.

## Author summary

The control of transcription is mediated in part through regulatory sequences that engage in a lineage and developmentally contextual manner. The *Golli/Mbp* locus gives rise to

**Funding:** This work was funded by the Canadian Institutes of Health Research, https://cihr-irsc.gc.ca/e/193.html (to KS and AP). The funders had no role in study design, data collection and analysis, decision to publish, or preparation of the manuscript.

**Competing interests:** The authors have declared that no competing interests exist.

several mRNAs and while *Mbp* mRNAs accumulate exclusively in the two glial cell types that elaborate myelin, *Golli* mRNAs accumulate in diverse cell types both within and beyond the nervous system. To determine how the different *Golli/Mbp* enhancers distribute their activities, and to reveal if they operate as autonomous agents or engage in functionally significant interactions, we derived multiple enhancer knock-out lines. Comparing the developmental accumulation of *Mbp* and *Golli* mRNAs revealed that the autonomous targeting capacity of multiple enhancers accurately predicted their *in situ* contributions where they acted in a largely additive manner accommodated by a putative chromatin-looping model. We also uncovered cryptic "stealth" activity that emanated from these same enhancers in lineages where they showed no autonomous targeting capacity. Thus, an unexpected and novel contribution from the super-enhancer domain also may be indicated.

## Introduction

Myelin facilitates rapid and energy efficient action potential conduction and when disrupted, as in demyelinating diseases such as Multiple Sclerosis, nervous system function can be severely compromised. In the mouse, myelination initiates between birth and weaning in both the CNS and PNS with peak accumulation of the mRNAs encoding myelin-specific proteins reached during the third postnatal week [1, 2]. Adaptive changes in myelin volume also occur in the mature nervous system where they are thought to influence circuit properties [3, 4]. Consequently, the mechanisms controlling myelin synthesis, including regulation of the genes encoding myelin related proteins, are the focus of intense investigation [5].

Myelin basic protein (MBP), a major component of myelin, is an intrinsically disordered protein susceptible to multiple post-translational modifications. While MBP is largely concentrated in the myelin sheath, it is implicated in a wide array of cellular functions [6]. Shiverer mice have a deletion in 3' domain of the *Mbp* locus, lack MBP and are incapable of elaborating compact CNS myelin [7–9]. In contrast, only limited ultrastructural anomalies and a modest decrease in myelin sheath thickness are observed in their PNS [10–13]. Notably, a previously reported positive correlation between the accumulated level of *Mbp* mRNA and CNS myelin sheath thickness demonstrated that MBP is not only essential but is a limiting factor in CNS myelin production [14, 15]. The *Golli/Mbp* locus also expresses *Golli* transcripts that initiate far upstream and incorporate various *Mbp* exons. Golli accumulates in diverse lineages both within and beyond the nervous system on cell-type specific developmental programs and has been shown to modulate $Ca^{2+}$ transients [16–22].

Many of the transcription factors (TFs) specifying myelinating cell lineages and/or controlling lineage maturation have been identified [5, 23–38]. Further, chromatin remodeling and epigenetic changes associated with myelin gene expression have revealed numerous features of the landscape in which such regulatory components operate [33, 34, 39–44]. Most notably, the *Mbp* enhancers investigated here have been shown to reside within a super-enhancer domain providing an environment thought to support exceptionally high concentrations of transcription factors and the intimate association of regulatory components in distinct condensates [45–47].

Motivated by the critical role myelin plays in nervous system function and the essential and rate-limiting role of MBP in CNS myelin formation, we, and others, have characterized features of the mechanism controlling transcription of the *Golli/Mbp* locus [1, 2, 26, 33, 34, 48–53]. Because *Mbp* is expressed by both myelinating cell types, accumulates in a well

characterized post-natal developmental program and has an unusual association with the widely expressed overlapping *Golli* transcriptional unit, the *Golli/Mbp* locus represents a particularly rich target within which any higher order organization of transcriptional regulators might be revealed.

Previously, the lineage specificities and developmental programs conferred by *Golli/Mbp*-associated enhancers were assigned using reporter constructs [2, 48, 50, 51, 53–55]. However, in such preparations, enhancers are isolated from their normal chromatin environment and often ligated adjacent to each other or directly to a promoter creating novel spatial relationships that may impose, or diminish, higher-order interactions [56]. Therefore, we sought to characterize enhancer contributions in the fully integrated context of the endogenous *Golli/Mbp* locus. Using CRISPR-based gene editing we derived lines of mice bearing alleles deleted of one or more enhancers and these were assessed for *Mbp* and *Golli* mRNA accumulation (relative to *Gapdh* mRNA) at key stages of post-natal development. Each enhancer KO allele caused the greatest mRNA reduction in the lineage where the respective enhancer conferred autonomous targeting. Unexpectedly, some enhancer deletions also led to reduced expression in the lineage where they demonstrated no autonomous targeting capacity and we refer to such additional cryptic function as "stealth" activity. The extent to which the super-enhancer contributes to such activity remains to be determined. Finally, our observations suggest a model in which transcription factor mediated interactions give rise to chromatin looping that brings the *Golli* and *Mbp* promoters and their relevant enhancers into close proximity.

## Results

### Enhancer knockout (KO) lines

Five domains of high interspecies conservation (M1-M5) are located 5' of the *Mbp* start site within a putative super-enhancer domain (Fig 1A). In previous studies, M1 (which encompasses the M1E enhancer and the contiguous *Mbp* proximal promoter), M3 and M5 were each shown to drive expression in oligodendrocytes while M4 drove expression in Schwann cells. M2 demonstrated no autonomous activity in either lineage [2, 48–50, 53]. In the present investigation we derived mouse lines homozygous for alleles deleted individually of M3, M4 or M5 or deleted of M3/M5 or M1E/M3/M5 combined. Additionally, to explore the potential function of an enhancer subdomain, the M3(225)KO allele bearing a partial deletion of M3 shown previously to upregulate reporter gene activity in both oligodendrocytes and Schwann cells was derived [49, 50] (Fig 1B). At key stages of myelination, accumulation of *Mbp* mRNA in spinal cord and sciatic nerve was analyzed while accumulation of *Golli* mRNA was assessed in spinal cord, thymus and retina at multiple ages.

### *Mbp* mRNA accumulation in spinal cord oligodendrocytes

Relative accumulation of *Mbp* mRNA was measured by analyzing whole tissue homogenates of spinal cord (CNS) and sciatic nerve (PNS) for oligodendrocytes and Schwann cells respectively. Oligodendrocytes initiate expression of *Mbp* as a terminal maturation event coincident with myelin sheath elaboration that initiates on widely different schedules in different CNS domains [1]. Consequently, we restricted our analysis to the cervical spinal cord where myelination initiates perinatally and oligodendrocyte numbers remain constant from at least P10 through P30 [57]. Thus, a close relationship between the mRNA levels observed and that realized within individual oligodendrocytes was expected. Samples were obtained at P7, a stage of maturation when significant myelin elaboration in cervical spinal cord has occurred [1]; at P14, when myelin acquisition nears peak levels; at P21, when the levels of myelin protein

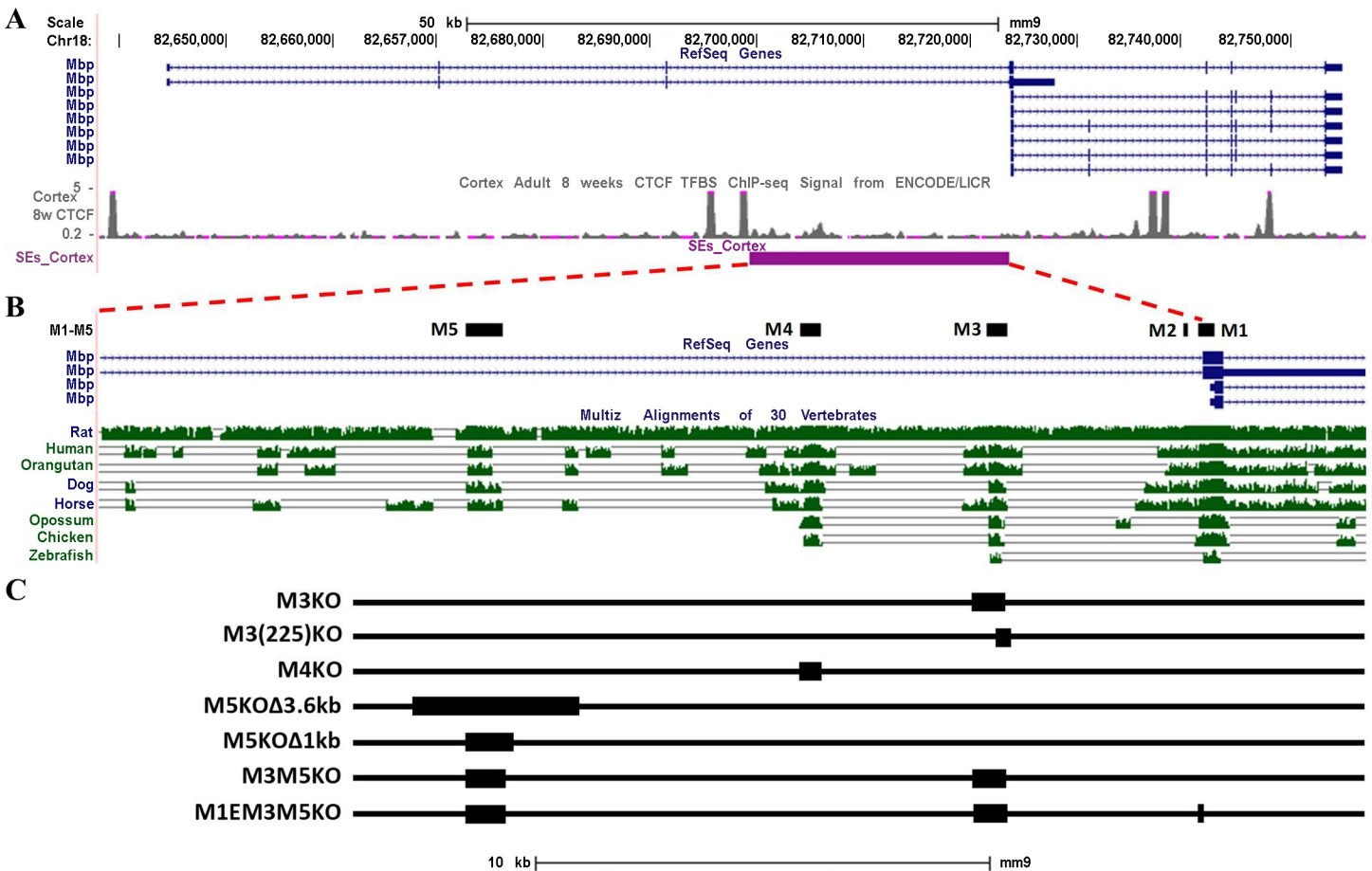

**Fig 1. Organization of the *Golli/Mbp* locus.** (A) *Golli/Mbp* locus with identified transcripts, CTCF binding signals and super-enhancer domain. (B) *Mbp* 5' flanking sequence indicating five modules of high interspecies sequence conservation in selected vertebrates with demonstrated autonomous enhancer activity (adapted from UCSC browser; see full species comparisons in mouse genome assembly NCBI37/mm9). Only representative *Golli/Mbp* transcripts are shown. (C) The position and length of individual and combined enhancer deletions derived into individual mouse lines.

mRNAs are declining from peak levels; at P30, when mature myelin maintaining cells predominate and at P90 when mice are fully mature.

**Wild type (WT).** In cervical spinal cord samples, *Mbp* mRNA was readily detectable at P7 and rapidly increased over the next week. Relative to P14, levels decreased to 85% by P21, 74% by P30 and 33% by P90; a developmental program consistent with prior investigations of multiple myelin gene expression programs and *Mbp* regulated reporters [1, 2, 48–50, 53, 58] (Fig 2, S1 Table).

**M5 KO lines.** M5 function has not been investigated in transgenic preparations but its capacity to drive transcription in the oligodendrocyte CG-4 cell line was detected using transfected reporter constructs [51]. Although the M5 enhancer displays only limited interspecies conservation, it is associated with a ChIP-Seq binding profile for myelin regulatory factor (MYRF) that extends for 917 bp encompassing a repeat domain. Three M5 deletions were generated; the 829 bp deletion (chr18: 82707436–82708264 NCBI/mm9) in the M3M5KO and M1EM3M5KO alleles removed only the conserved non-repeat sequence; the 1014 bp deletion (M5KOΔ1kb) (chr18: 82707433–82708446 NCBI/mm9) extends 185 bp further 3' to include a short repeat region, while the 3647 bp deletion (M5KOΔ3.6kb) (chr18: 82706381–82710027 NCBI/mm9) encompasses the conserved 829 bp as well as multiple flanking repeat domains. A

similar reduction in *Mbp* mRNA accumulation was observed with the M5KOΔ1kb and M5KOΔ3.6kb alleles at P14 demonstrating that the deleted repeat sequences do not function, at least at this age (S1 Table).

**M3KO, M5KOΔ3.6kb, M3M5 double KO and M3(225)KO.** Oligodendrocytes in mice bearing either M3KO (chr18: 82718631–82719324 NCBI/mm9) or M5KOΔ3.6kb alleles accumulated *Mbp* mRNA in similar developmental programs that differed only modestly at P7 and P90 when the M5KOΔ3.6kb allele supported relatively higher accumulation (Fig 2, S1 Table). Notably, while M3 and M5 share multiple TF binding peaks including those for Sox10 and OLIG2 [52, 59], they differ for others including MYRF, present only in M5 [51], and ZFP24, present only in M3 [36]. The combined M3M5KO allele resulted in an additive reduction in *Mbp* mRNA accumulation with only a modest difference observed at P7. Typical of WT, M3KO and M5KOΔ3.6kb mice, those bearing the M3M5KO or M1EM3M5KO alleles demonstrated an approximately 2-fold upregulation from P7 to P14. In contrast, the rapid downregulation realized by P21 did not occur in the M3M5KO or M1EM3M5KO lines in which only a limited drop off was observed at P90. Although, the partially truncated M3(225) allele supported massive and continuous upregulation of reporter gene expression [50], the identical *in situ* truncation in M3(225)KO (chr18: 82719120–82719395 NCBI/mm9) led to equivalent or even modestly reduced *Mbp* mRNA accumulation relative to M3KO.

**M1EM3M5 triple KO.** The *Mbp* proximal promoter sequence extending to -300 bp 5' of the transcription start site failed to drive reporter expression in oligodendrocytes [53]. However, a sequence extending further 5' to -377 bp (M1) supported transient expression during primary myelination thus demonstrating that the 5' 77 bp (referred to as the M1 enhancer,

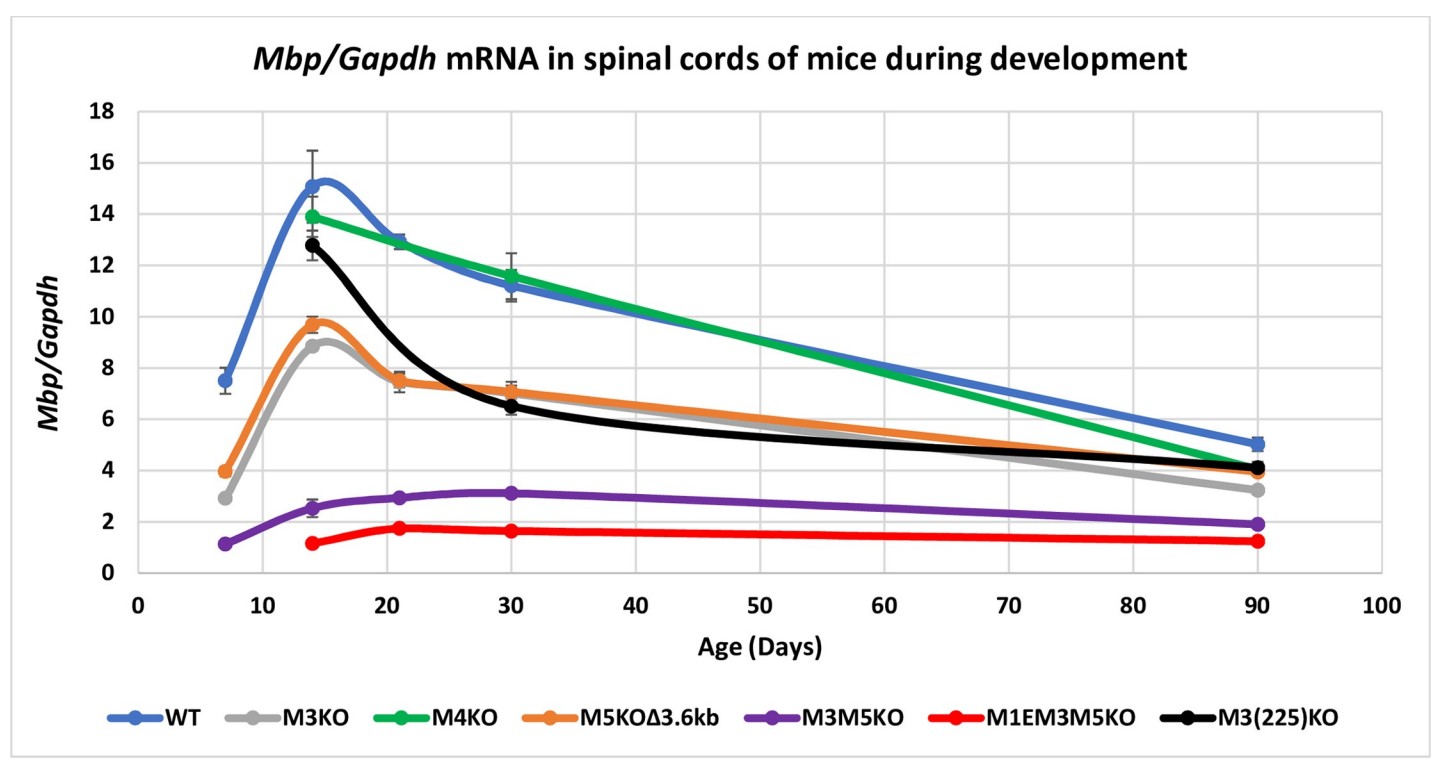

**Fig 2. Relative *Mbp* mRNA accumulation in cervical spinal cord.** M1E, M3 and M5 are major enhancers of *Mbp* in oligodendrocytes. X-axis = post-natal age in days and Y-axis = *Mbp/Gapdh* mRNA ratio. Dots represent values at P7, 14, 21, 30 and 90 and connecting lines are the predicted developmental programs calculated as trendlines (Excel). Error bars = SEM.

M1E) is required for oligodendrocyte-specific activity. The M1EM3M5 allele investigated here is deleted of M3 and M5 in addition to M1E (chr18: 82723587–82723666 NCBI/mm9) and the functional significance of M1E was inferred by comparison of the M1EM3M5KO and M3M5KO alleles. Notably, M1EM3M5KO mice demonstrated a developmental expression program that paralleled that of M3M5KO mice but at approximately half absolute levels (Fig 2, S1 Table). Consistent with previous models in which reduced *Mbp* levels were shown to limit CNS myelin production [14, 15, 60], the M3M5KO mice exhibited pronounced CNS hypomyelination (S1 Fig). Further, M1EM3M5KO mice that accumulated *Mbp* mRNA to approximately 14% of WT at P21 demonstrated a shivering phenotype similar to that reported previously for transgenic mice that accumulated *Mbp* mRNA to 13.5% of WT at P18 [14].

**M4KO.** M4 drives reporter expression only in Schwann cells and, consistent with that autonomous activity, the M4KO allele (chr18: 82714831–82715271 NCBI/mm9) had no effect on *Mbp* mRNA accumulation in oligodendrocytes at P14 or P30. However, significantly reduced accumulation, albeit modest, was observed in oligodendrocytes at P90 (Fig 2, S1 Table).

## *Mbp* mRNA accumulation in sciatic nerve Schwann cells

**WT and M4KO.** In WT sciatic nerve *Mbp* mRNA was readily detectable at P4, reached a peak level at P14 and declined to 42% of the P14 value by P90. In previous reporter-based investigations, *Mbp* 5' sequence extending to -8.9 kb (thus encompassing M1-M3) drove expression in oligodendrocytes but remained silent in Schwann cells. In contrast, 5' sequence extending to -9.4 kb, and thus incorporating the full M4 conserved sequence, drove robust expression in both oligodendrocytes and Schwann cells [49, 53]. Consistent with the strong Schwann cell specific programming observed with all M4 bearing reporter constructs, M4KO mice accumulated *Mbp* mRNA at much reduced levels, approximating 20% of WT at all ages (Fig 3, S2 Table).

**M3KO, M5KOΔ3.6kb, M3M5KO, M1EM3M5KO and M3(225)KO.** In contrast to all M4 bearing reporter constructs, no M3 constructs driven through the *Mbp* proximal promoter (M1) expressed in Schwann cells. However, when M3 was ligated to a heterologous and minimal *hsp* promoter, transient Schwann cell expression was observed during a restricted period of preweaning myelin elaboration [53]. Thus, while M3 is not capable of productively engaging the *Mbp* proximal promoter in Schwann cells, it nonetheless must bind relevant TFs. Consistent with that interpretation, sciatic nerve demonstrates a binding peak for Sox10 and enrichment for H3K27ac over M3 [52]. Unexpectedly, all lines bearing an allele deleted of M3 demonstrated reduced *Mbp* mRNA accumulation in Schwann cells throughout development as did those bearing the M5KOΔ3.6kb allele, although in a more modest fashion. Further, the reduction seen in mice bearing the combined M3M5KO allele trended lower than either the M3KO or M5KOΔ3.6kb alleles at most ages and was strictly additive at P14. The accumulation program in M1EM3M5KO mice was comparable to that observed in M3M5KO mice while the partially deleted M3(225)KO allele had no effect (Fig 3, S2 Table).

## *Golli* mRNA accumulation

In the mouse, *Golli* transcription initiates approximately 80 kb upstream of *Mbp* and *Golli* isoforms variously incorporate specific *Mbp* exons as well as 264 bp of the *Mbp* proximal promoter [16, 61]. The M1-M5 *Mbp* regulatory modules are located in the 20 kb 5' of the *Mbp* start site and therefore within a *Golli* intron. In the oligodendrocyte lineage, *Golli* accumulates predominantly in progenitors and is observed only rarely in post-mitotic myelinating oligodendrocytes [62]. However, *Golli* is expressed by numerous other cell types, both within and beyond the nervous system, including neurons and T-cells where distinct lineage specific

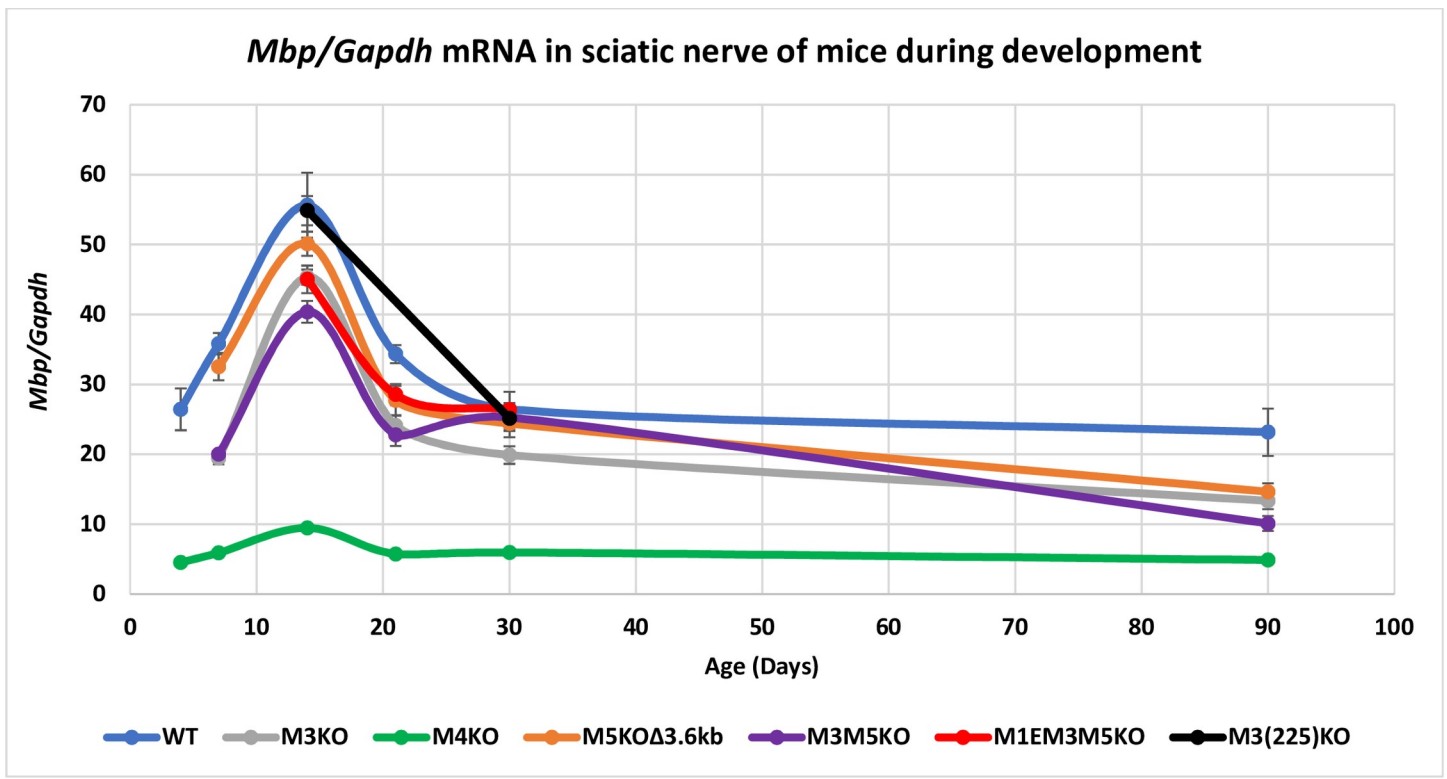

**Fig 3. Relative *Mbp* mRNA accumulation in sciatic nerves.** M4 is the major *Mbp* enhancer in Schwann cells. Loss of M3 and/or M5 alone resulted in lower accumulation at several ages while loss of M1E has no additional affect. X-axis = age in days and Y-axis = *Mbp/Gapdh* mRNA ratio. Dots represent post-natal days P4, 7, 14, 21, 30 and 90 and connecting lines are the predicted developmental programs calculated as trendlines (Excel). Error bars = SEM.

developmental programs are observed [16, 62–66]. Therefore, the *Golli* mRNA values obtained from spinal cord likely arise from a combination of cell types that potentially changes with development. To determine levels of expression in nervous tissue devoid of oligodendrocytes we examined retina and for T-cells we examined thymus. Expression in Schwann cells was not determined as *Golli* accumulation was at the limit of detectability in sciatic nerve samples. Consistent with previous findings for optic nerve [49], *Golli* mRNA accumulation was reduced to approximately 10% of WT in all tissues examined from the lines in which M3 was deleted. The absence of the M1E sequence in the M1EM3M5KO allele had no further effect. In contrast, M5KOΔ3.6kb and M4KO lines demonstrated normal *Golli* mRNA accumulation in all tissues examined. Notably, in samples from M3(225)KO mice, *Golli* accumulation was normal at P14, indicating that the specific M3 subsequence deleted in this allele plays no role in *Golli* regulation at this age. However, at P30, *Golli* accumulation increased to 141% and 130% of WT in spinal cord and thymus respectively, indicating an age-specific putative repressor role for the deleted sequence (Figs 4 and 5, S3 Table). However, that relationship was reversed at P90 where accumulation reached only 65% of WT.

## Discussion

In this investigation we sought to define the developmental programming conferred by different *Mbp* enhancers in the context of the endogenous *Golli/Mbp* locus. We expected such insight to lay the functional foundation upon which investigations capable of revealing mechanistic insights into myelin gene programming can be realized. Further, by contrasting such

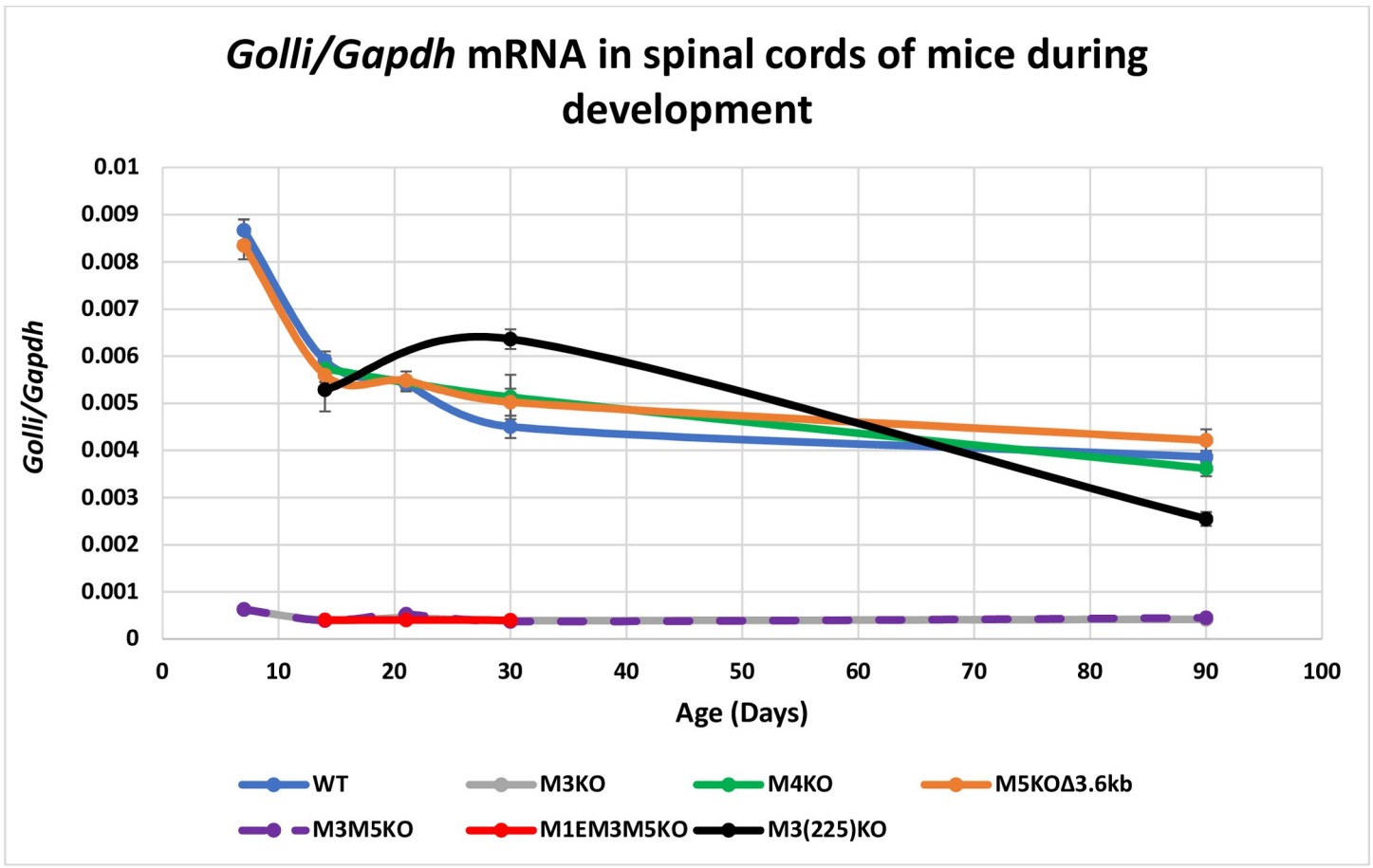

**Fig 4. M3 is a major *Golli* enhancer in spinal cord.** All alleles deleted of M3 demonstrate an approximately 90% reduction in *Golli* mRNA accumulation in spinal cord. In contrast, the partially deleted M3(225)KO allele supported accumulation ranging from a mild decrease to a modest increase at different ages. X-axis = age in days and Y-axis = *Golli/Gapdh* mRNA ratio. Dots represent post-natal days P7, 14, 21, 30 and 90 and connecting lines (both solid and dashed) are the predicted developmental programs calculated as trendlines (Excel). Error bars = SEM.

integrated activity with that realized through enhancers isolated in reporter constructs, we expected any higher order levels of functional interaction to be exposed. As *Mbp* expression follows a tightly scripted *in vivo* developmental program, we evaluated mRNA accumulation in both CNS and PNS at key stages of primary myelin elaboration through to the mature myelin maintenance phase.

Extensive characterization of the autonomous activity conferred by M1, M3 and M4 was achieved previously using randomly inserted reporter constructs [1, 2]. In all cases, multiple expressing transgenic lines bearing different copy number inserts yielded similar lineage specificities. Subsequently, those initial investigations were complemented with controlled transgenic preparations in which single copy constructs were inserted in a predetermined orientation at a common and permissive chromatin site [48–50, 53]. The latter approach also supported direct comparisons between the quantitative outputs conferred by different enhancer sub-sequences. Reporters driven through the *Mbp* promoter bearing all enhancer combinations investigated revealed identical targeting capacities in both types of transgenic preparations. These observations provided the required basis from which the autonomous regulatory capacity of enhancers could be compared with high confidence to their contributions in the context of the endogenous *Golli/Mbp* locus.

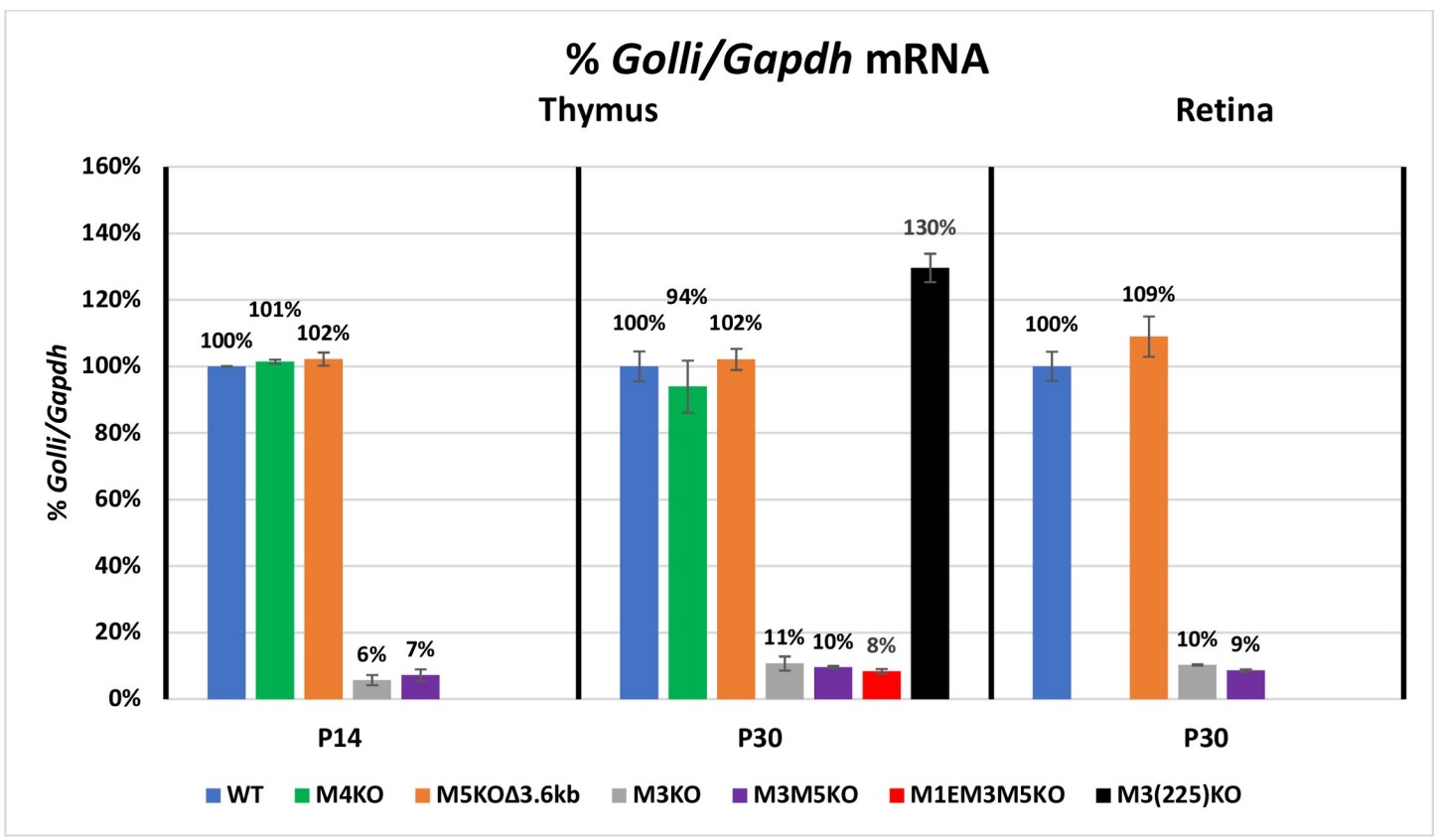

**Fig 5. M3 is also the major *Golli* enhancer in thymus and retina.** All alleles deleted of M3, but not the partially deleted M3(225)KO allele, reduced *Golli* accumulation to a similar extent. X-axis = age in days and Y-axis = %*Golli/Gapdh* mRNA. Error bars = SEM.

Here, mRNA analysis was performed on whole tissue samples thus making the assay susceptible to multiple potential variables including changes in the density and/or differentiation state of relevant cell populations, compensatory feedback mechanisms and secondary changes beyond the transcriptional level. However, these potential caveats are largely dispelled by previous observations. Firstly, shiverer mice, despite their *Mbp* null status and fully amyelinated CNS, have a normal density of mature oligodendrocytes from P10 through P30, although a notable increase is observed at P60 [57]. Consequently, in enhancer KO lines that demonstrate only reduced *Mbp* mRNA accumulation, rather than a total absence, oligodendrocyte density is expected to be normal at least through P30; the period in which the majority of the present observations were made. However, whether an increase in oligodendrocyte density occurs by P90 remains to be determined. In a similar manner, the accumulation of other myelin gene products in the amyelinating oligodendrocytes of *Mbp* null shiverer mice, including the prominent *Plp* transcript and protein, show little change from WT [67, 68]. Further, the rate of initiation of their truncated *Mbp* transcript also tracks closely that of WT [69]. Thus, changes in cell density or major disruptions in myelin gene programming were not anticipated responses to the reduced *Mbp* mRNA accumulations imposed by the enhancer KOs. A similar relationship was expected to exist in the PNS; e.g., by P5 in WT mice, spinal root axons are heavily myelinated and Schwann cell proliferation has ceased [10]. Further, shiverer mice demonstrate only minor ultrastructural anomalies and mild thinning of PNS myelin sheaths [10, 13]. Finally, the endogenous *Mbp* accumulation program observed in enhancer KO mice is reflected in the reporter accumulation programs observed in many transgenic lines that bear

an intact *Golli/Mbp* allele [1, 2, 48–50]. Therefore, we interpret observations made on the different KO lines to reflect a valid estimate of the average *Mbp* mRNA accumulation realized by generally equivalent populations of oligodendrocytes and Schwann cells.

## Expression of reporter genes vs. enhancer KOs

Extensive prior investigations exploiting reporter constructs revealed multiple features of *Mbp* enhancer function extending to their autonomous lineage specificities and functional capacities of their sub-sequences. However, reporter constructs are unlikely to reveal the full extent of the integrated regulatory activity conferred by individual enhancers as insertion site, copy number and multiple levels of chromatin organization, beyond that associated with the endogenous *Golli/Mbp* locus, could influence transcriptional efficiency. A notable level of such higher-order organization is the super-enhancer, a chromatin domain encompassing one or more transcriptionally active enhancers frequently associated with key lineage-specific or lineage specifying genes [54, 70–73]. Super-enhancers are enriched in active chromatin modifications, bind master TFs, mediator and co-activators and their boundaries are often demarcated by duplicated CTCF sites. Association between CTCF binding factor and cohesion supports the formation of chromatin loops creating insulated enhancer bearing neighborhoods [70]. Further, recent evidence shows that super-enhancer components, brought into close proximity, can form liquid-like condensates in which transcriptional components are highly enriched [45–47].

Multiple data sets from human brain revealed a super-enhancer domain extending through and 35 kb upstream of the *Mbp* gene [52, 71]. Mouse cortical samples revealed a similar domain of 24 kb enriched in active transcription marks such as H3K27ac and H3K4me1 and demarcated by paired CTCF binding sites within which the M1-M5 enhancers are located [55, 74, 75]. In the CNS, M1, M3 and M5 (but not M4) demonstrate H3K27ac enrichment and are bound by SOX10, a glial specifying transcriptional activator with a potential role in regulating the formation of super-enhancer domains [52]. Further, SOX10 has been shown to interact with mediator in both oligodendrocytes and Schwann cells [29]. Whether this super-enhancer domain forms exclusively in *Mbp*-expressing oligodendrocytes or also in Schwann cells remains to be determined. Nonetheless, in Schwann cells, the M1, M3, M4 and M5 enhancers are all enriched in H3K27ac and bind SOX10 [34, 52]. Finally, multiple *Golli*-expressing tissues demonstrate variably sized super-enhancer domains elsewhere in *Golli/Mbp* locus.

The extent to which super-enhancers confer synergy on their constituent enhancers remains controversial [46, 47, 71, 75–79]. Should inter-enhancer interactions occur within the *Golli/Mbp* super-enhancer, we expected them to be revealed by comparing the outputs of different enhancer KO alleles. In contrast to the synergy model, in oligodendrocytes the double M3M5KO allele gave rise to the precise reduction of *Mbp* mRNA predicted by combining the individual consequences of the M3 and M5 KO alleles. Such observations are consistent with an additive model in which super-enhancer activity equals the sum of its constituent enhancer activities.

In multiple reporter configurations, the M4 enhancer revealed strong and autonomous targeting activity only in Schwann cells [2, 48, 49, 53]. In contrast, all contiguous 5' sequences terminating before M4, but encompassing M3, failed to drive reporter expression in Schwann cells. As predicted by these observations, the M4KO allele resulted in a major reduction in *Mbp* mRNA accumulation in Schwann cells (~20% of WT). However, despite the failure of M3 to support reporter expression in Schwann cells, its absence in KO lines caused a significant reduction in *Mbp* mRNA accumulation in that lineage at all ages examined. We identify this additional cryptic activity arising from the otherwise autonomous oligodendrocyte specific

M3 enhancer as "stealth" activity. Defined in this manner, stealth activity also was exhibited in oligodendrocytes by the otherwise autonomous lineage specific M4 Schwann cell enhancer, albeit only at P90. Although the reduction in mRNA accumulation observed in M5KOΔ3.6kb Schwan cells may reflect similar stealth activity, this cannot be confirmed as the autonomous *in vivo* targeting capacity of M5 is yet to be investigated.

We are not aware of previous experiments designed to reveal stealth enhancer activity such as that assigned here to M3 and M4. Thus, it remains to be determined if the stealth phenomenon is widespread, unique to *Golli/Mbp* or limited to enhancers within a common super-enhancer domain. However, that certain enhancers affect activity only through association with special "hub" enhancers has experimental support [76, 80]. M3, and possibly M5, might participate in Schwann cells as non-hub enhancers engaging with M4, while in oligodendrocytes, M4 might act as a non-hub enhancer engaging with any of the oligodendrocyte enhancers. Notably, M3 and M5 both show H3K27ac enrichment and Sox10 binding in the PNS while M4 binds OLIG2 in immature oligodendrocytes [52, 59]. Unique to the previously hidden non-autonomous stealth activity observed here is its origin from enhancers that drive robust fully autonomous expression only in a different lineage.

## Reporter genes vs. partial enhancer module KOs

Insight into the location of TF binding and chromatin modifications associated with the sequences engaged in *Mbp* enhancer activity has been obtained from prior functional and ChIP-Seq analysis [33, 34, 36, 51, 52, 59] (S4 Table). Complementing such investigations have been constitutive and conditional KO of the TFs and chromatin modifiers themselves [24, 26, 32, 37, 42, 59, 81–89]. However, KOs of oligodendrocyte or Schwann cell TFs often disrupt their maturation prior to expression of *Mbp* and myelin formation and/or their survival and cell identity thus making it difficult to assign a direct role for such TFs to *Mbp* transcription per se. An interesting example is MYRF, a presumptive master regulatory TF that, within the super-enhancer, only binds to M5 [51]. In *Myrf* KO mice, maturation of oligodendrocyte is arrested affecting the expression of multiple myelin genes and preventing *Mbp* expression [27]. However, the KO of M5 did not similarly prevent *Mbp* expression but resulted in only a ~40% reduction in its accumulation. Here we contribute to this analysis by investigating the function of alleles bearing partial deletions of M1 and M3. Absence of the 77 bp M1E sequence in the M1EM3M5KO allele reduced *Mbp* mRNA accumulation in oligodendrocytes beyond that imposed by the M3M5KO allele alone, demonstrating an enhancing role for the M1E domain (Fig 2). In contrast, loss of M1E had no effect on *Mbp* mRNA accumulation in Schwann cells, demonstrating that it confers neither enhancing activity nor plays any role in the productive engagement of M4 with the remaining promoter components in that lineage (Fig 3).

When M3 was ligated to M1 promoted reporters it failed to drive expression in Schwann cells. However, when ligated to a minimal *hsp* promoter it conferred transient pre-weaning reporter expression. In marked contrast, the partially deleted M3(225) sequence conferred constitutive expression at levels up to 50 fold higher in Schwann cells and up to 5 fold higher in oligodendrocytes [50]. In the context of the endogenous locus, the same M3 truncation (M3 (225)KO) paradoxically had no effect on *Mbp* mRNA accumulation in Schwann cells and, relative to WT, reduced rather than enhanced accumulation in oligodendrocytes (Figs 2 and 3). The basis for this striking difference between reporter construct activity and that realized by the same deletion in the endogenous locus remains unknown but the experimentally imposed close proximity of M3(225) and the *hsp* promoter in reporter constructs may contribute to the observed upregulation.

Further demonstration that distinct enhancer sub-domains affect promoter-enhancer relationships was revealed by *Golli* expression. All alleles deleted of M3 reduced *Golli* mRNA accumulation to approximately 10% of WT in all samples examined (Figs 4 and 5). In marked contrast, the partially truncated M3(225)KO allele supported near normal *Golli* mRNA accumulation in spinal cord at P14, modest upregulation at P30 and only mild downregulation at P90. Notably, the upregulation at P30 was observed also in thymus. Thus, M3 sub-sequences differ in their capacity to engage the *Mbp* and *Golli* promoters, consistent with a model in which M3 functions through different TFs as a general "house-keeping" enhancer for *Golli* and a strong lineage-specific enhancer for *Mb*p.

## A model accommodating enhancer targeting activities

The *Golli/Mbp* enhancer activities, and their functional interactions documented here, lead to a DNA-looping model compatible with much of the observed integrated output of the locus (Fig 6A and 6B). This model accommodates previously described differential TF binding by different *Mbp* enhancers [32, 46, 70, 90–94]. Specifically, self-associating TF dimers, such as those formed by YY1 and SP1 are implicated in chromatin-looping and long-distance enhancer-promoter interactions [56, 95–99]. This mechanism could accommodate the emergence of multiple regulatory programs such that a single enhancer can regulate two different promoters and the output of a single locus can evolve to match the unique requirements of the different cell types that myelinate the CNS and PNS.

M1, M3, M4, M5 and the *Golli* proximal promoter all have predicted high affinity YY1 binding sites, while only M3, M3(225) and the *Golli* proximal promoter have similarly high affinity SP1 binding sites (Jaspar; relative score (rs) > 90%) [100]. However, lower affinity SP1 and YY1 motifs (80% < rs < 90%) exist in all. Accordingly, in oligodendrocytes, M3 and M5 interaction with the *Mbp* promoter could involve DNA-looping mediated by YY1 and/or SP1 dimerization (Fig 6A). While conditional KO of YY1 in oligodendrocytes leads to amyelination characteristic of MBP null shiverer mice, post-mitotic oligodendrocyte maturation unfortunately is blocked such that any direct role for YYI in *Mbp* transcription cannot be easily revealed in that model [88, 89]. Notably, the M3(225)KO allele lacks the YYI motif and exhibits the same reduction seen in mice bearing the full M3KO mice at P30. However, higher levels of *Mbp* mRNA accumulation are observed at P14 and P90 and this partial restoration of enhancer-promoter interaction might be conferred through its retained SP1 motif (Fig 6A). These observations suggest that age-specific differences in capacity to promote inter-sequence interaction may exist. Indeed, the global methylation of DNA in both oligodendrocytes and Schwann cells is known to change during differentiation and myelination [101] and YY1 DNA binding is methylation sensitive [102]. In addition, during myelin synthesis in oligodendrocytes SP1 becomes phosphorylated via PKC/Erk [103, 104]; a pathway shown to increase its DNA binding capacity in smooth muscle [105]. During this period, SP1 accumulation and binding to the *Mbp* promoter also increases [103, 104].

Activity mediated in part by SP1 and/or YY1 dimerization provides a convenient model that appears to accommodate many of our observations on *Mbp* expression and an expansion of this looping model may also accommodate *Golli* programming (Fig 6B). While the *Golli* proximal promoter contains both YY1 and SP1 binding motifs, among the enhancers investigated here, only the M3 enhancer modulates *Golli* output and it uniquely contains a high affinity SP1 binding site [100]. Moreover, the truncated M3(225)KO allele that retains this SP1 binding site drives high *Golli* expression at all ages (Fig 6B).

Beyond insight into the functional organization of *Golli/Mbp* regulatory sequences, further aspects of oligodendrocyte biology are revealed by the developmental programs realized by the

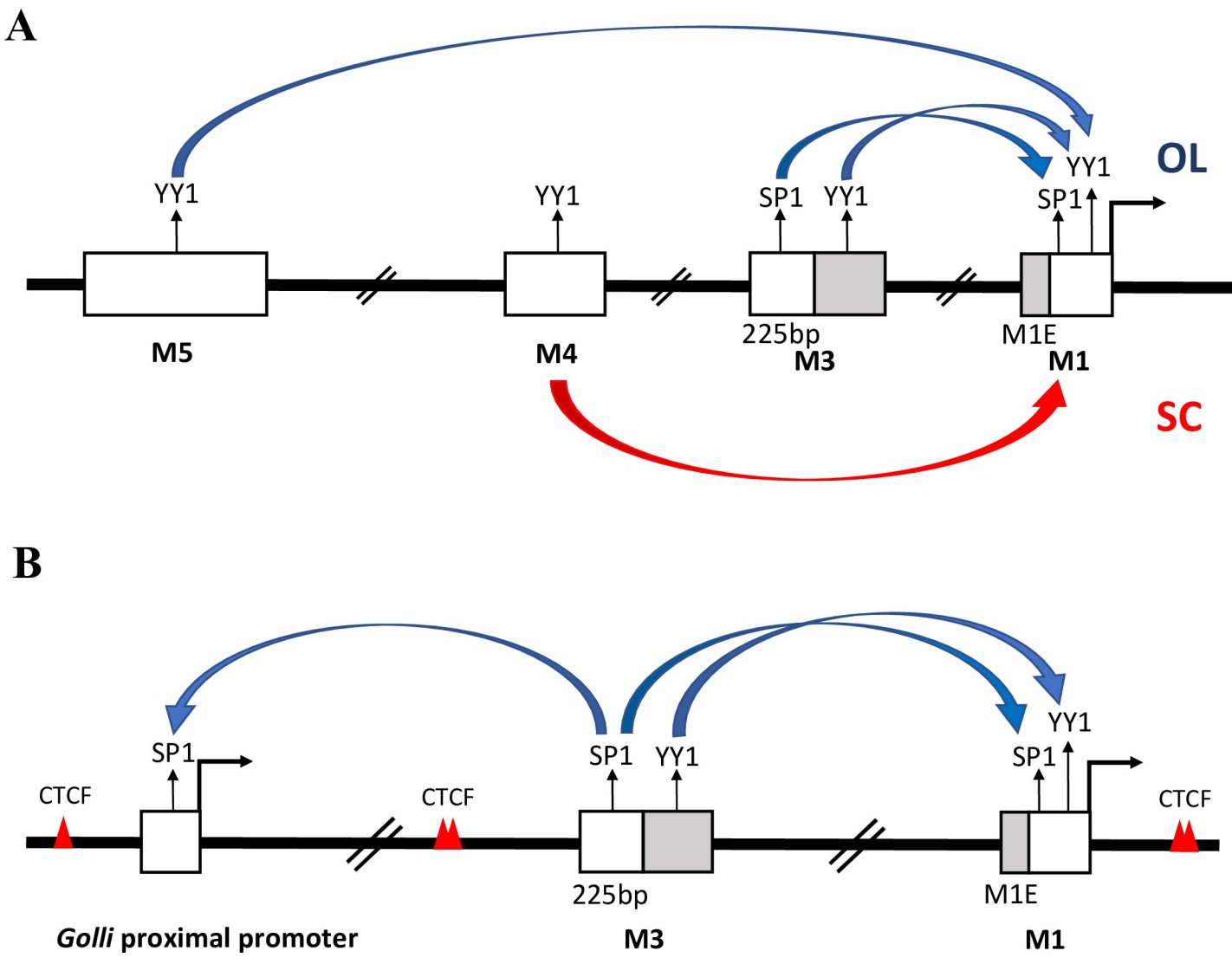

**Fig 6. A tentative DNA-looping model for enhancer engagement.** (A) The predicted direct enhancer-promoter interaction for *Mbp* expression in oligodendrocytes (top with blue arrows) and Schwann cells (bellow with the red arrow). (B) The predicted relationship between M3 and the *Mbp* and *Golli* promoters. White boxes = enhancers and proximal promoters. Deleted sequences are highlighted in grey. Arrows = potential interactions. Red triangles = CTCF binding sites.

endogenous *Mbp* locus, relevant reporter constructs and the KO alleles reported here. As demonstrated by the capacity of oligodendrocytes to myelinate inanimate fibers *in vitro*, initial myelin elaboration can be supported entirely by oligodendrocyte intrinsic programming [106]. In contrast, it is widely recognized that myelin in the mature CNS demonstrates plastic changes potentially in response to neuronal activity [3, 4, 107, 108]. Consistent with such developmental changes, numerous reporter constructs demonstrated maturation specific expression programs as observed here with the M3(225)KO and M4KO alleles [48, 50]. These observations are consistent with a model in which the qualitative and/or quantitative features of the transcription factor repertoire evolves during maturation; a circumstance indicating the eventual necessity to evaluate mechanisms regulating myelin gene transcription in the *in vivo* developmental context.

Like WT mice, all enhancer KO lines showed an approximately 2-fold increase in *Mbp* mRNA accumulation in spinal cord between P7 and P14. Similar upregulation was observed in sciatic nerve samples. Notably, in the brains of amyelinated shiverer mice, expression of their truncated *Mbp* transcript increases in a similar manner during this period [69]. An increased density of oligodendrocytes or Schwann cells may contribute to that rise but if so, such contributions are expected to be small; the density of mature oligodendrocytes in spinal cord is unchanged between P10 and P30 [57] and Schwann cell proliferation terminates by P5 in spinal roots when myelination is well advanced [10]. However, regardless of what combination of enhancers are deleted, an early and seemingly uniform developmental rise is observed.

A rapid and pronounced decline in *Mbp* mRNA accumulation occurs in WT mice prior to weaning. However, this decline was not uniform in the enhancer KO lines. While M3KO and M5KOΔ3.6kb lines followed the WT program, the double M3M5KO and triple M1EM3M5KO lines did not. Rather, their accumulation levels at P14 remained unchanged through P30 after which they reduced only slightly. What accounts for this striking difference remains unknown but our combined evidence suggests that at least two different programs regulate normal *Mbp* expression; one responsible for its upregulation during the primary myelination period and another that maintains expression at maturity [3, 108]. Indeed, this is consistent with observations from reporter studies in which regulatory sequences were shown to have different expression capacities in young vs. old mice [48, 50].

While the present study characterizes the major regulatory sequences encompassed within the oligodendrocyte super-enhancer, other potential regulatory sequences located elsewhere in the *Golli/Mbp* locus have emerged from ChIP-Seq analysis and these may contribute to the residual expression observed in the enhancer KO lines (S4 Table).

This investigation provides insights into the complex regulatory mechanisms governing *Golli/Mbp* programming and lays the required functional framework from which the role of chromatin configuration and modification along with specific TF binding can be approached [76]. As the mouse models described here contain unique configurations of *Golli/Mbp* regulatory sequence they may themselves contribute to such investigations including the formation and function of super-enhancers and their potential role in stealth activity. This study further suggests the future requirement to assess the transcriptional mechanisms controlling myelin genes at key stages of *in vivo* maturation. Finally, while these enhancer-deleted mouse models are fully viable, they elaborate myelin sheaths of variably reduced thickness providing a unique opportunity to re-evaluate basic features of the axon-myelin relationship.

## Materials and methods

### Animals

All experiments were carried out in accordance with the guidelines of the Canadian Council on Animal Care. Protocol number 215–7668 approved by the McGill University DOW Facilities Animal Care Committee.

### CRISPR design and gene editing

**M1E, M4 and M5 sequences.** The 422bp M4 enhancer targeted here was described previously [48]. M5 refers to the target of the MYRF ChIP carried out in rat [51]. Using the UCSC browser, this sequence was aligned with the mouse genome (chr18: 82707095–82708011 NCBI/mm9). For the purpose of generating M4 and M5 enhancer KOs, single guide RNAs (sgRNAs) were designed to target sequences flanking the conserved enhancer domain such that double strand breaks would be simultaneously introduced at both sides of the enhancer

resulting in deletion of the intervening enhancer sequence. sgRNAs were designed (using the CRISPR Design http://crispr.mit.edu/) [109] to identify locus specific targets. To minimize the potential impact of inefficient sgRNAs, we designed 2 that bind in close proximity (for a total of 4 sgRNAs per enhancer deletion). The sgRNA target sequences used to generate the KO mice are indicated in S5 Table.

**sgRNA design.** The plasmid DR274 was a gift from Keith Joung (Addgene plasmid # 42250) [110]. DR274 was digested with BsaI which cuts twice between the T7 promoter and the gRNA scaffold, leaving sticky ends. For each of the targets listed above, two oligos, one for each strand, were ordered from IDT (S6 Table). They were annealed at 40uM each in NEB3 buffer. Each has one of the DR274 sticky ends so that they could be ligated into the plasmid using the NEB Quick Ligation Kit (M2200S). Each ligation mix was transformed into competent bacteria and kanamycin resistant clones obtained. 3 clones of each were sequenced in the relevant region and used to generate the sgRNA. To generate the sgRNA template for the M4 deletion, the PCR method using two long overlapping oligos was used (S6 Table) [111]. The MEGA shortscript T7 kit from Life Technologies was used to synthesize the sgRNA from the T7 promoter. The resulting sgRNA was tested for integrity on a Bioanlyzer at the McGill Genome Center.

The target-specific crRNAs (S6 Table) were hybridized to Alt-R® CRISPR-Cas9 tracrRNA, the Universal 67mer tracrRNA from IDT to generate the functional sgRNAs according to the manufacturer's instructions. AltR1 and AltR2 are proprietary (IDT) modifications to increase the stability of these short RNAs.

## Zygote manipulation, delivery of CRISPR components and transplantation into pseudopregnant mice

Zygotes were recovered mid-day from the oviduct of WT or M3KO C57Bl/6 mice [49] naturally mated to wmN2 transgenic mice [112]. The cumulus cells were removed by a short incubation in 1% hyaluronidase/M2 medium (Millipore) and moved into Advanced KSOM media (Millipore) at 37°C with 5% $CO_2$. All zygote manipulation was done at room temperature and the media was kept under mineral oil. M4KO mice were generated by microinjection into zygote cytoplasm of 25ng/ul Cas9 mRNA (PNA Bio) and 12.5ng/ul of each of four sgRNAs. All M5 deletions were generated by zygote electroporation. Prior to electroporation the zygotes were moved to Opti-MEM (Life Technologies) and thinning of the zona was achieved by treating the zygotes with Acid Tyrode's solution (Millipore) for 10 seconds and transferring them back into fresh Opti-MEM. Zygotes were electroporated according to the ZEN2 protocol [113, 114] with a final concentration of 250ng/ul Cas9 mRNA (PNA Bio) and 300ng/ul sgRNA dissolved in TE pH7.5/Opti-MEM at a 1:1 ratio [113]. A 20ul drop of this mix containing the CRISPR reagents was prepared and the batch of 30–50 zygotes carried in less than 1ul of Opti-MEM were moved into this drop. The mix was transferred to a 1 mm electroporation cuvette purchased from BioRad and electroporation was carried out using a Bio-Rad Gene Pulser Xcell electroporator. Embryos were subjected to 1–2 pulses of 25–30 V according to the ZEN protocol [113]. After microinjection or electroporation embryos were cultured overnight in advanced KSOM media at 37°C with 5% $CO_2$. After overnight incubation, embryos at the 2-cell stage were transplanted (bilaterally, approximately 15/mouse) into the fallopian tubes of CD1 female recipients rendered pseudopregnant by mating with B6C3F1 vasectomized males (purchased from Charles River).

## Genotyping and breeding scheme

Pups were tail-biopsied at weaning for genotyping. Tail samples were digested at 55C overnight in lysis buffer (containing 100 mM Tris, pH 8.0, 5 mM EDTA, pH8.0, 200 mM NaCl,

0.2% sodium dodecyl sulfate (SDS) and 100ug/ul proteinase K) and genomic DNA was extracted. Genotyping initially was done using PCR with primers surrounding the sequence to be deleted. Upon detection of a desired, shorter-than-WT, band, the PCR product was sequenced at the McGill University and Génome Québec Innovation Centre and the existence of M4 and M5 deletions confirmed. Founder mice were mated to WT C57Bl/6 and the consequent progeny were genotyped by PCR for the deletion and LacZ (to detect the presence of a transgene at the HPRT locus, that exists within our donor colony and select against it). Mice carrying the enhancer deletion were mated to homozygosity while breeding out the transgene located on the X chromosome. In total, 2 lines of M4KO mice (identical sequencing results), 2 lines bearing single M5KOs (different deletion lengths), 1 line of M3M5 double KO and 1 line of M1EM3M5 triple KO were established.

## Tissue samples

After the homozygous lines of mice were established, samples from 3–11 mice of both genders from WT and all KO lines were obtained at the ages indicated (S1, S2 and S3 Tables). The mice were anesthetized with a lethal dosage of Avertin and sciatic nerve, cervical spinal cord, retina and thymus samples were collected into RNAlater solution (Ambion) according to the manufacturer's instructions and stored at -20˚C.

## RNA extraction and qRT-PCR

Total RNA extraction was done using Trizol (Life Technologies) and a Qiagen RNeasy MinElute Cleanup kit. RNA was eluted in nuclease free water and its concentration was measured using a spectrophotometer. The RT reaction was carried out using Superscript IV VILO Mastermix (Life Technologies) using 1ug of total RNA according to the manufacturer's instructions and the resulting cDNA was stored at -80˚C. A QuantStudio™ 7 Flex Real-Time PCR System (Life technologies) was used for qPCR in a 96-well plate. On the day of qPCR, the cDNA was diluted 20x and 40x for measuring *Golli* and 100x and 400x for measuring *Mbp* and *Gapdh*. Each sample was measured twice at the low dilution and once at the high dilution. To avoid inter-plate variability samples from WT and KO mice were measured together on individual 96-well plates. To measure *Mbp* and *Gapdh* in SN and cervical spinal cord, Taqman probes (*Mbp*: Mm01266402_m1, *Gapdh*: Mm99999915_g1, Life technologies) were used. For *Golli* measurements in cervical spinal cord and thymus however, the SYBR green method was used (PowerUp SYBR green master mix, Life Technologies). Multiple primer sets were designed, tested and the optimal pair (2F: 5'ATTGGGTCGCCATGGGAAAC, 2B: 5'CCAGC CTCTCCTCGGTGAAT) was chosen. On each plate, 5 10-fold serial dilutions of a DNA standard were run in triplicate to generate a standard curve. Standards were prepared by amplifying a sequence larger than the measured amplicon. After standard PCR, the single band was purified from a gel with a NucleoSpin Gel and PCR cleanup kit (Macherey-Nagel) and its concentration determined. The efficiencies of reactions for both Taqman and SYBR green methods inferred from standard curves were 95–105%.

## Data analysis

After qRT-PCR, sample measurements from multiple dilutions were averaged. Relative amounts of *Mbp* and *Golli* were calculated by dividing the average of each by the average of *Gapdh* for the same sample. The relative *Mbp* and *Golli* measurements of all samples of each mouse line were averaged and the standard error of the mean was calculated. To determine statistical significance, ANOVA with post-hoc Tukey test was performed using IBM SPSS

Statistics 25 software. As analysis of sub-samples revealed no gender differences, male and female samples were combined throughout this investigation.

## Light and electron microscopy

Mice lethally anesthetized with Avertin, were transcardially perfused with 2.5% glutaraldehyde + 0.5% paraformaldehyde in 0.1M sodium cacodylate buffer and cervical spinal cord and ON samples were collected. Samples were postfixed overnight at 4°C followed by rinsing with 0.1M sodium cacodylate buffer. A second post fixation was done with 1% osmium tetroxide followed by rinsing with ddH2O. Samples were dehydrated by incubation in increasing concentrations of acetone: 30%, 50%, 70%, 80%. 90% and 3X100%. Infiltration was done with 1:1, 2:1, 3:1 (epon:acetone) followed by embedding in epon and overnight polymerization at 60°C. 0.5 um sections were stained with Toluidine blue and cover slips were mounted with epon for imaging. Slides were imaged with 63X or 100X oil immersion objectives by light microscopy (Zeiss Axio Imager M1).

## Supporting information

**S1 Fig. Mice bearing the M3M5KO allele demonstrate CNS hypomyelination.** Electron micrographs were obtained from cross sections of the ventral medial cervical spinal cord from P90 WT and M3M5 mice (**A** and **B** at 640x) and (**C** and **D** at 3000x). The axon population in this domain ranges from small to large calibers. In the M3M5KO sample, axons of all calibers are typically ensheathed with compact myelin markedly thinner than normal although rare small calibre axons lacking compact myelin (*) were encountered.
(PDF)

**S1 Table. Relative *Mbp* mRNA analysis in spinal cord of enhancer knock-out mice at P7, P14, P21, P30 and P90.** The values are presented as % ± standard error of the mean. "*" and "**" represent p-values $\leq 0.05$ and $\leq 0.01$ respectively. n(F:M) represents the number of Female and Male mice from each genotype analyzed at each age.
(PDF)

**S2 Table. Relative *Mbp* mRNA analysis in sciatic nerve of enhancer knock-out mice at P4, P7, P14, P21, P30 and P90.** The values are presented as % ± standard error of the mean. "*" and "**" represent p-values $\leq 0.05$ and $\leq 0.01$ respectively. n(F:M) represents the number of Female and Male mice from each genotype analyzed at each age. "#" indicates that sciatic nerves from two mice were combined for each sample.
(PDF)

**S3 Table. Relative *Golli* mRNA accumulation in spinal cord of enhancer knock-out mice at P7, P14, P21, P30 and P90.** The values are presented as % ± standard error of the mean. "**" represent p-values $\leq 0.01$. n(F:M) represents number of Female and Male mice analyzed.
(PDF)

**S4 Table. Oligodendrocyte and Schwann cell ChIP-Seq data relevant to the *Golli/Mbp* locus.** References are indicated by ().
(PDF)

**S5 Table. sgRNA target sequences used to generate the KO mice.**
(PDF)

**S6 Table. Oligonucleotides used for CRISPR editing.**
(PDF)

## Acknowledgments

McGill facilities: Facility for Electron Microscopy Research (FEMR). McGill University and Genome Quebec Innovation Centre. We thank Drs. E. Cooper, K. Dewar and X-J Yang, McGill University, for helpful comments.

## Author Contributions

**Conceptualization:** Hooman Bagheri, Hana Friedman, Katherine A. Siminovitch, Alan C. Peterson.

**Data curation:** Hooman Bagheri.

**Funding acquisition:** Katherine A. Siminovitch, Alan C. Peterson.

**Investigation:** Hooman Bagheri, Hana Friedman, Alan C. Peterson.

**Methodology:** Hooman Bagheri, Hana Friedman, Alan C. Peterson.

**Project administration:** Katherine A. Siminovitch, Alan C. Peterson.

**Supervision:** Alan C. Peterson.

**Writing – original draft:** Hooman Bagheri, Hana Friedman, Katherine A. Siminovitch, Alan C. Peterson.

**Writing – review & editing:** Hooman Bagheri, Hana Friedman, Katherine A. Siminovitch, Alan C. Peterson.

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
