## [Decision Letter · Decision Letter 0]

26 Apr 2020

Dear Dr Peterson,

Thank you very much for submitting your Research Article entitled 'TRANSCRIPTIONAL REGULATORS OF THE GOLLI/MYELIN BASIC PROTEIN LOCUS INTEGRATE ADDITIVE AND STEALTH ACTIVITIES' to PLOS Genetics. Your manuscript was fully evaluated at the editorial level and by independent peer reviewers. The reviewers appreciated the attention to an important problem, but raised some substantial concerns about the current manuscript. Based on the reviews, we will not be able to accept this version of the manuscript, but we would be willing to review again a much-revised version. We cannot, of course, promise publication at that time.

As you will see, some of the reviewers noted the lack of evidence to substantiate the interesting models you propose, and were concerned that without additional experimental support, your conclusions were mostly speculative.  Some reviewers suggested removing some of the conclusions that they felt were particularly speculative given the available data; however, simply removing the unsupported claims will not be sufficient as a response to these reviews. 

While all reviewers agreed that the submitted manuscript represents a monumental amount of work with great potential for broad relevance, in the absence of data -either published data from other groups or results from new experiments- to support your conclusions and to provide some mechanistic insights, the present manuscript will fall short of the broad appeal and level of advance that is expected for a paper in PLoS Genetics.  The reviewers provide some suggestions for published data you might incorporate, and suggest additional experiments as well.

If you decide to revise the manuscript for further consideration at PLOS Genetics, please aim to resubmit within the next 60 days, unless it will take extra time to address the concerns of the reviewers, in which case we would appreciate an expected resubmission date by email to plosgenetics@plos.org.

[LINK]

We are sorry that we cannot be more positive about your manuscript at this stage. Please do not hesitate to contact us if you have any concerns or questions.

Yours sincerely,

Lisa Stubbs

Associate Editor

PLOS Genetics

Gregory Barsh

Editor-in-Chief

PLOS Genetics

Reviewer's Responses to Questions

**Comments to the Authors:**

Reviewer #1: The manuscript by Bagheri and colleagues is on the characterization of regulatory elements within the Golli/Mbp locus of mice. Using CRISPR/Cas9-dependent genome editing the authors remove single sequence elements that have been previously characterized in trangenic mice as oligodendrocyte- or Schwann cell-specific enhancers and together form a super-enhancer. The authors find that the single enhancer elements exhibit the cell-specific enhancer activity previously detected in standard transgenesis and can determine the relative contribution of each single enhancer to overall activity. In addition to this, there are a few bits of interesting additional information. For one, the authors find stealth activity for a Schwann cell-specific enhancer in oligodendrocytes and vice versa. Second, they find additive rather than synergistic activity of single enhancers in the super-enhancer structure. Finally, they observe a discrepancy in the activity of an enhancer truncation in standard transgenesis and CRISPR/Cas9-dependent genome editing.

The manuscript is well written and the data are clearly presented. The analysis on enhancer activity almost exclusively relies on RT-PCR of bulk RNA from spinal cord, sciatic nerve, thymus and retina, and is therefore unable to analyze whether changes of expression are homogeneous or heterogenous between cells of the same type. There is also the possibility that reduced expression in the normally expressing cell type is (partially) counteracted by ectopic expression generated as a consequence of genome editing. However, considering the wealth of information on this particular genomic locus, chances are very high that the authors‘ interpretations of RT-PCR on bulk RNA are correct.

To me the strength oft he manuscript is that it summarizes a huge amount of mouse work and represents a heroic effort of settling once and for all the enhancer landscape oft he Golli/Mbp locus. The obvious drawback of the manuscript is that it does not go beyond a standard expression analysis of the various enhancers. There is, for instance, no attempt to provide experimental evidence for the looping model provided in Fig. 6. Taken everything together, I find the work highly informative and expect much added value to the field of glial biology. Therefore I am in favour of publication.

Reviewer #2: This manuscript reports an in vivo characterization of multiple enhancers for the mouse Golli/Mbp locus. The mouse Golli/Mbp locus derives two transcripts, Golli and Mbp mRNA, by two alternative promoters. Yet, the expression of Golli transcripts are ubiquitous, whereas the expression of Mbp transcripts is limited to nerve cells. According to previous studies, the transcription of this locus is regulated through the 5 enhancers (named M1 through M5) that are localized in the 20-kb upstream region of Mbp promoter, which are also the 3rd intron in the case of Golli transcripts. In this study, the authors have generated a series of deletion KO models targeting these enhancers, and subsequently characterized in vivo functions of these enhancers. They measured the expression levels of Mbp mRNA using the total RNA isolated from spinal cord and sciatic nerves, representing the oligodendrocytes and Schwann cells, respectively. They also performed a similar series of qRT-PCR analyses for Golli mRNA with the total RNA isolated form thymus and retina cells. According to the results, M3 and M5 enhancers are critical for the expression of Mbp in the oligodendrocytes, M4 enhancer for the expression of Mbp in the Schwann cells, and finally M3 enhancer for the expression of Golli in all the tissues tested. Besides these main observations, they also observed functional contributions by M3 and M5 enhancers to the expression of Mbp in the Schwann cells. This was unexpected, since M3 and M5 were previously shown to be non-functional in this nerve cell. Given these observations, the authors suggest that M3 and M5 enhancers may have a hidden or ‘stealth’ activity for the transcription of Mbp in the Schwann cells. Also, the authors proposed a model involving various chromatin loop structures to explain how one enhancer (M3) could be involved in two different expression patterns: lineage-specific expression for Mbp transcripts and ubiquitous expression for Golli transcripts.

Overall, this study provides several new insights regarding the in vivo functions of several enhancers for the mouse Golli/Mbp locus. However, the authors need to consider the following criticisms to improve their manuscript for future publication.

First, this manuscript seems to have a big, but unsubstantiated, jump from the conclusions from experimental results to the subsequent interpretations or claims. As described earlier, the immediate conclusions are simple and straightforward, which are confirming the predicted tissue-specific functions of three enhancers, M3, M4, and M5. Yet, the main claims include 1) the hidden or stealth activity of M3 and M5 in the unexpected cell type, the Schwann cells, and 2) potential interactions of the individual enhancers for the lineage-specific and ubiquitous expression of Mbp and Golli mRNA, respectively. As part of their claims, the authors propose a model involving chromatin loop formations, but without any actual data, such as 3C or other related results. They further suggest potential involvement of Sp1 and Yy1 transcription factors for these chromatin loop formations, but again without any actual demonstration of the binding of these transcription factors to this mouse locus. The authors need to first perform additional experiments and derive the relevant results to propose their model.

Second, the authors need to provide a more detailed description of the mouse Golli/Mbp locus, showing the exon structures of Golli and Mbp mRNA along with the positions of primer sets for qRT-PCR as part of Figure 1. According to the publically available data from UCSC genome browser, this locus is much larger than the one shown in Figure 1, and, furthermore, the upstream region and the other two introns of Golli, 1st and 2nd introns, seem to have many other potential enhancers that have never been mentioned in the manuscript. These are also likely involved in the transcriptional regulation of this locus, thus need to be discussed along with the 5 known enhancers.

Third, there are many run-on sentences throughout the text of this manuscript, which are very confusing. These long sentences need to be broken down properly for the better reading of the audience.

Reviewer #3: The regulation of the Mbp gene in oligodendrocyte biology has been the focus of many studies in the last couple decades. While transcription factors regulating Mbp have been defined, there has been relatively little analysis of which enhancers are required at the endogenous locus for induction and maintenance of Mbp expression. The manuscript describes a series of mouse lines in which genome editing was used to delete enhancers of the Mbp gene. Some of the enhancers had been shown to regulate expression of this myelin-associated gene in both oligodendrocytes and Schwann cells, but previous experiments had relied on transgenic reporter assays. The advance here is a combinatorial and comprehensive analysis of five Mbp enhancers using genome editing to create enhancer knockouts. The studies show the enhancer contributions to developmental regulation in both oligodendrocytes and Schwann cells along with regulation of the golli transcript. The analysis shows additive contributions of some specific enhancers in the oligodendrocyte lineage (including one activated by Myrf) and the predominance of one Mbp enhancer in Schwann cells. Relatively few previous papers have used genome editing to remove oligodendrocyte enhancers at endogenous loci, although one enhancer deletion (M3) had been previously described.

The analysis is carefully done and represents a significant amount of effort. The developmental time course analysis provides important data during the active period of myelination and also during myelin maturity.

One technical concern is that some of the deletions cause hypomyelination due to fairly low Mbp levels, and there is some concern that changed expression levels may be due indirectly to changes in oligodendrocyte differentiation upon Mbp depletion. It would be strongly recommended to assess this by looking at other oligodendrocyte genes in the time course (e.g. Plp1, Cnp, Myrf) to assess this possibility. This issue should be considered in light of results of Aaker et al, ASN Neuro 2016, which should be cited and discussed.

Discussion Points:

1. The manuscript should provide a more comprehensive summary and integration of previous data on transcription factors that regulate Mbp and any available published ChIP-seq data for the five enhancers. One notable omission is Zfp24 which binds to M3 (Elbaz et al., 2018), and ChIP-seq data are also available for Olig2. In addition, how does the decrease in Mbp expression observed by deleting the Myrf-bound M5 enhancer correlate with the decreased Mbp expression in the Myrf knockout? Since Myrf might have been thought to be required for Mbp induction, some explanation should be provided for the relatively modest effect of M5 deletion. Some Mbp regulators like Sox10/Olig2 are thought to be active throughout OPC/oligo development, while others like Myrf and Zfp24 are activated in myelinating oligos, so some mechanistic synthesis would be useful for the field.

2. The results are considered in the context of a looping model, which is appropriate although 3C data for Mbp in this cell type is not available. However, more recent super enhancer studies have incorporated the condensate model of enhancer activity, which depends upon enhancer recruitment of a critical mass of factors that can induce partitioning of genes into transcriptionally active condensates. The Discussion can be revised to consider alternate models such as this.

3. The specification of enhancers having "stealth" activity perhaps should be reconsidered as there does not have appear to be precise mechanistic definition of this property, other than enhancer activity was not detected in previous transgenic assays. Historically, some enhancers do not show activity unless multimerized so the absence of activity in a given transgenic context could depend on limitations of the original transgenic design.

Reviewer #4: Bagheri et al. provide an in vivo functional dissection of the enhancers in the Golli/Myelin Basic protein locus in oligodendrocytes and Schwann cells from transgenic mice. Different enhancers in this locus were deleted by CRISPR-Cas9 in mouse and its effects of the expression of MBP and Golli in vivo in oligodendrocytes and Schwann cells was tested at different stages of mouse development. Different conclusions on the workings of these enhancers, that together form a super enhancer, were drawn.

The study provides interesting information on the regulation of an important locus. The conclusions are generally supported by the data and give important information that could not be drawn from earlier reporter studies of these enhancers. The results show mostly additive effects of deletions of individual enhancers, which adds to the discussion whether super enhancers as to whether super enhancers should be viewed as one functional unit or more as a collection of closely spaced but autonomously acting enhancers.

I support publication of this study in principle, but a few issues have to be addressed first.

The authors have to be more clear about their deletions. M5KOF6 should be also in Fig. 2. The M5KO line has a much bigger deletion in M5 than the M3M5 KO and M1EM3M5 KO line. The M5 deletion in the latter two lines is never tested in unison and its effect is therefore unknown. It would be good if the authors mark the different M5 deletions in their figures so the reader understands that they are different. It would also be good if the authors mention the number of mice used per experiment also in the materials and methods.

The conclusion that the M5 multiple repeat regions do not influence function is not tested and should be removed. For this, one has to compare M5 deletions with and without (all) repeat regions and this was not done, for reasons unknown as it would make the experiments more clean.

On the discussion, I agree that M3 has stealth activity in Schwann cells. However, the reduction MBP expression in M4 KO in oligodendrocytes is simply too few to conclude that M4 has stealth activity in oligodendrocytes. This conclusion should be removed. M5 KO has a small effect on MBP expression in Schwann cells but as its reporter activity was not tested, if I understand correctly, the hint that it also has stealth activity should be removed.

**Have all data underlying the figures and results presented in the manuscript been provided?**

Reviewer #1: Yes

Reviewer #2: Yes

Reviewer #3: No:

Reviewer #4: Yes

PLOS authors have the option to publish the peer review history of their article (what does this mean?). If published, this will include your full peer review and any attached files.

Reviewer #1: No

Reviewer #2: No

Reviewer #3: No

Reviewer #4: Yes: Raymond A. Poot

---

## [Editor Report · Decision Letter 1]

1 Jul 2020

Dear Dr Peterson,

We are pleased to inform you that your manuscript entitled "TRANSCRIPTIONAL REGULATORS OF THE GOLLI/MYELIN BASIC PROTEIN LOCUS INTEGRATE ADDITIVE AND STEALTH ACTIVITIES" has been editorially accepted for publication in PLOS Genetics. Congratulations!

Yours sincerely,

Lisa Stubbs

Associate Editor

PLOS Genetics

Gregory Barsh

Editor-in-Chief

PLOS Genetics

Comments from the reviewers (if applicable):

**Data Deposition**

http://datadryad.org/submit?journalID=pgenetics&manu=PGENETICS-D-20-00449R1

**Press Queries**

---

## [Editor Report · Acceptance letter]

4 Aug 2020

PGENETICS-D-20-00449R1 

TRANSCRIPTIONAL REGULATORS OF THE GOLLI/MYELIN BASIC PROTEIN LOCUS INTEGRATE ADDITIVE AND STEALTH ACTIVITIES 

Dear Dr Peterson, 

We are pleased to inform you that your manuscript entitled "TRANSCRIPTIONAL REGULATORS OF THE GOLLI/MYELIN BASIC PROTEIN LOCUS INTEGRATE ADDITIVE AND STEALTH ACTIVITIES" has been formally accepted for publication in PLOS Genetics! Your manuscript is now with our production department and you will be notified of the publication date in due course.

With kind regards,

Kaitlin Butler

PLOS Genetics

On behalf of:
